# Overview of Neo-Vascular Lesions after Delivery or Miscarriage

**DOI:** 10.3390/jcm10051084

**Published:** 2021-03-05

**Authors:** Yuji Shiina

**Affiliations:** Department of Obstetrics and Gynecology, Yamagata Prefectural Shinjo Hospital, Shinjo, 12-55 Wakabacho, Yamagata 996-0025, Japan; shiina@mail.dewa.or.jp; Tel.: +81-233-22-5525; Fax: +81-233-29-2693

**Keywords:** placental polyps, retained products of conception, arteriovenous malformations, neo-vascular lesion

## Abstract

The concept of intrauterine neo-vascular lesions after pregnancy, initially called placental polyps, has changed gradually. Now, based on diagnostic imaging, such lesions are defined as retained products of conception (RPOC) with vascularization. The lesions appear after delivery or miscarriage, and they are accompanied by frequent abundant vascularization in the myometrium attached to the remnant. Many of these vascular lesions have been reported to resolve spontaneously within a few months. Acquired arteriovenous malformations (AVMs) must be considered in the differential diagnosis of RPOC with vascularization. AVMs are errors of morphogenesis. The lesions start to be constructed at the time of placenta formation. These lesions do not show spontaneous regression. Although these two lesions are recognized as neo-vascular lesions, neo-vascular lesions on imaging may represent conditions other than these two lesions (e.g., peritrophoblastic flow, uterine artery pseudoaneurysm, and villous-derived malignancies). Detecting vasculature at the placenta–myometrium interface and classifying vascular diseases according to hemodynamics in the remnant would facilitate the development of specific treatments.

## 1. Introduction

The sudden onset of critical vaginal bleeding 12 years after pregnancy was reported in 1884 in Philadelphia [1]. The disease, so-called placental polyps, was first recognized in this case report. Further details of this critical bleeding disease after delivery or miscarriage, the criteria for its diagnosis, and its etiology have gradually accumulated [2,3,4,5].

Placental polyps occur in <0.25% of pregnancies [3,5]. Furthermore, only 6% of placental polyps are hypervascular and cause severe hemorrhage, and the term hypervascular placental polypoid mass (HPPM) is sometimes used [6]. Placental polyps are strictly defined pathologically only by the examination of resected specimens. A placental polyp contains predominantly ghost villi that are hyalinized and necrotic, without lining trophoblasts. Some of the chorionic villi show a rim of viable syncytiotrophoblasts. The syncytiotrophoblasts may contribute to stimulating neovascularization in the myometrium. Partial hyalinization of the vascularization around the villi may lead to hemorrhage [6].

Remnants of placenta or a membrane attached to the uterine wall and fibrin deposition around the remnants are usually considered pathognomonic of the formation of placental polyps. The development of these remnants, that is, the pathogenesis of the placental polyps, has been explained by two theories. According to the first theory, the cornual or fundal site of the uterine myometrium is thin and atonic, so that the placental tissue attached to this region is easily retained. The second theory is that of placenta accreta, in which the villi are directly attached to the underlying myometrium due to the defective decidua, easily leading to the retention of the placental tissue combined with uterine atony [3,5]. Placental tissue attaches at the fundus or cornu, and then the trophoblastic villi invade the myometrium at the site of attachment. Placental polyps are of two types. The first type occurs in the first four weeks after the postparturition period, and the other one occurs months or years later [5]. The factors responsible for the survival of these villi are still unclear.

## 2. New Concept and Diagnosis

A piece of retained placenta may eventually form so-called placental polyps. Retained pieces of placental tissue are common causes of bleeding. These retained fragments refer to placental tissue retained in the uterine cavity, and clinical bleeding after abortion or full-term delivery is now initially evaluated by diagnostic imaging, e.g., ultrasonography (US) and magnetic resonance imaging (MRI). The disease concept has thus recently shifted from a pathological definition to one based on diagnostic imaging. The new diagnosis of retained products of conception (RPOC), which is now used widely in clinical practice, is currently defined as the presence of a mass considered to be derived from residual placenta or villi after delivery or miscarriage, sometimes with blood flow within this residual structure. Color Doppler images are used “subjectively” to diagnose a rich vascular network. In addition, the peak systolic velocity (PSV) is used “objectively” to review the outline of the disease’s natural history [7,8,9]. According to the new concept based on imaging, these patients’ findings are diagnosed as RPOC clinically [4,10].

The characteristics of RPOC are different from those of placental polyps, which are diagnosed by pathological examination. Placental polyps were previously considered to occur in 1 in 40,000–60,000 pregnancies and to be very rare [5]; based on diagnostic imaging, however, the number of case reports is increasing. Previously, the main reports were of critical hemorrhage, but a recent report stated that the lesions may regress, and spontaneous recovery may occur within a few months in many cases [7,11,12]. There are three issues that may have given rise to these differences. The first is that the disease concepts of placental polyps and RPOC are not completely consistent; the second is that blood flow is sometimes present and sometimes absent in RPOC; and the third is that neo-vascular lesions include disorders other than RPOC. In the vast majority of clinical settings today, diagnostic imaging is used to make this diagnosis, but the mixing of the old and new disease concepts is causing confusion. The aim here is to clarify these two.

## 3. Relationship between RPOC and Placental Polyps

RPOC is a condition that starts at miscarriage or delivery, and the residual placenta pieces or villi change their pathophysiology over time. RPOC is more than just remnants: it is a condition that develops secondary changes with the passage of time. RPOC initially consists of retained villi. In the next stage, the residual villi undergo necrosis with the deposition of fibrin, producing the pathological condition called placental polyps. Placental polyps may thus be viewed as a specific form of RPOC constituting an extreme expression of this condition. The patient presents with severe hemorrhaging when the polyp is detached [10].

In clinical practice, the issue is the severity of genital bleeding. Even persistent bleeding can be treated electively with conservative therapy if the amount of hemorrhage is small, but critical bleeding may require hysterectomy. The evaluation of vascular lesions can be useful for predicting such a hemorrhage. However, the form of placental polyps that corresponds to vascular lesions has yet to be defined. There is also a need to redefine placental polyps in a form other than pathological diagnosis, in the context of the series of changes over time in RPOC. The only option is to carry out imaging evaluations of vascular lesions that appear likely to cause critical hemorrhage and to regard them as corresponding to the lesions formerly referred to as placental polyps. Placental polyps are an extreme form of RPOC, but there are limits to their evaluation by imaging. The next section considers RPOC with vascularization.

## 4. Neo-Vascular Lesions of RPOC

Neo-vascular lesions may arise from remnant placenta or villi. Lesions appear and are accompanied by frequent abundant vascularization in the myometrium attached to the remnant. One characteristic of neo-vascular lesions in RPOC is that they grow toward the myometrium. These phenomena can involve residual villi undergoing necrosis, the formation of arteriovenous fistulas, and continuing development of arteriovenous communication [10]. These neo-vascular lesions were not originally present but developed over time after miscarriage or delivery. They are benign neoplasms and go through proliferating and involuting phases. Many of these vascular lesions have been reported to resolve spontaneously within a few months [13]. Although careful monitoring for sudden hemorrhage is required, their potential for spontaneous regression makes them candidates for elective treatment [12].

According to a report [14], MRI findings of RPOC basically include a combination of three parts in varying degrees: presence of a remnant, breaking of the junctional zone in contact with the remnant, and vascularization. The absence of the junctional zone on MRI suggests that placenta accreta is the basis of the pathogenesis. At the defective portion of the decidua, the villi and the myometrium directly contact each other, resulting in various morbid conditions arising in the villi of this direct adhesion site [4,5]. How vascularization develops from the defective part in the decidua basalis may form the basis for a clinically useful classification system. With accurate classification of the vascular lesions, it is possible to develop treatment protocols tailored to each type of RPOC.

## 5. Neo-Vascular Lesions of AVMs

Should a hypervascular appearance with turbulent flow on color Doppler ultrasound be evident after delivery or miscarriage, this may be a completely different condition from RPOC. Close investigation shows that vascularization on imaging includes conditions other than RPOC. The differential diagnosis of vascular lesions of the uterus must also include acquired arteriovenous malformations (AVMs) [15]. Basically, AVMs are congenital lesions, but sometimes acquired lesions develop. Acquired AVMs may result from causes including pelvic surgery, trauma, curettage, trophoblastic disease, diethylstilbestrol (DES) exposure, neoplasm, or infection. AVMs appear as a hypervascular mass at the point of contact between the remnant placenta or villi and the myometrium. In some cases, no remnant is evident, and only the vascular lesion is present. In AVMs, the neo-vascular lesion does not necessarily grow toward the myometrium, and they are characterized by hypervascularization within the AVM. These neo-vascular lesions start to be constructed during the formation of the placenta; they do not develop over time following miscarriage or delivery. AVMs are errors of morphogenesis with stable cellularity—defects in the structure of an organ—and do not show spontaneous regression [16]. Since there is no possibility of their spontaneous regression, they must be treated immediately. During a critical hemorrhage, hysterectomy must be performed to save the patient’s life, but when the hemorrhage is less severe and more time is available, uterine artery embolization (UAE) is an option to spare the uterus [17,18,19].

Angiography, which is the gold standard for the definitive diagnosis of AVMs, can identify the feeding arteries and draining veins of the AVM, which are characterized by an arterial phase with piles of newly engorged vessels and a dilated venous phase that is immediately visualized. In some cases, the arterial and venous phases may be visualized simultaneously, indicating the presence of an arteriovenous shunt. AVMs can be definitively diagnosed by the evaluation of these plexi [20], but this examination is uncommon clinically. Angiography is not available for initial screening at all times. MRI may be helpful as a second-best option. On MRI, as described above, the condition arises from the combination of three factors: the presence of a remnant, breaking of the junctional zone in contact with the remnant, and vascularization [14]. The direction in which vascularization is progressing may be particularly helpful in differentiating these lesions from RPOC.

## 6. Differentiation between RPOC with Vascularization and an AVM (Case Report)

We treated a patient with suspected RPOC with vascularization that was, in fact, an AVM. We describe this case below. This case involved a 28-year-old, gravida 1 para 0 woman whose pregnancy progressed without any particular complications. On day 281 of pregnancy, the spontaneous delivery of a 3175-g girl occurred. The placenta was expelled and showed no apparent signs of a defect 13 min postpartum. Massive bleeding occurred suddenly 2 h postpartum, with blood loss since delivery totaling 1271 g. The administration of a uterotonic in combination with intravaginal gauze packing successfully stopped the bleeding. Transabdominal ultrasonography showed an intrauterine mass measuring 45 × 35 mm^2^, raising suspicion of RPOC. Ultrasound showed a hypervascular appearance with turbulent flow on color Doppler, and this was initially thought to be typical RPOC with vascularization. On close observation, however, the hypervascularity was limited to the point of contact between the remnant and the myometrium, and there was no pulsatile perfusion in the remnant itself (Figure 1) [19,21].

Angiography demonstrated abnormal engorgement in the arterial phase, with a thick venous phase visualized only 2 s later. This indicated the presence of an arteriovenous shunt, and an AVM with a hypervascular region was diagnosed (Figure 2) [20].

There was no pulsatile arterial perfusion in the remnant itself, where the contrast agent simply appeared to be leaking. The remnant was not the primary locus of the disease, but it was rather acting as a cover preventing major bleeding from the AVM. This case was correctly diagnosed as an AVM rather than RPOC with vascularization. UAE was performed, after which the remnant detached spontaneously, and the uterus was preserved.

Currently, in clinical practice, a hypervascular appearance with turbulent flow on color Doppler ultrasound after delivery or miscarriage is sometimes observed roughly, and RPOC with vascularization is not clearly distinguished from AVM. This may result in the performance of unnecessary interventions for RPOC when hemorrhage is unlikely to occur or the mistaken choice of elective therapy for AVMs. Reaching a clear differential diagnosis leads to the right treatment strategy. It enables a clear policy to be decided: should a patient be treated electively and conservatively, or would proactive intervention be better [12]? Careful observation of the state of perfusion is required to make the differential diagnosis.

## 7. Neo-Vascular Lesions of Other Conditions

Occasionally, a hypervascular appearance with turbulent flow is also observed in normal pregnancy due to vasodilatation and decreased resistance in the spiral arteries. Such “peritrophoblastic flow” appears in the myometrium in the very early intrauterine gestation, even before a gestational sac is visible, or following a recent spontaneous or induced abortion [22].

A uterine artery pseudoaneurysm is one of the conditions of intrauterine vascularization after delivery or miscarriage [23]. A pseudoaneurysm, which forms adjacent to an artery, is a “pool” of blood, so to speak, since it lacks a vascular wall structure and is only enclosed by surrounding tissues [24]. Characteristic features include an intrauterine hypoechoic mass on ultrasound and the yin-yang sign on color Doppler ultrasound [25]. Sometimes, subsequent spontaneous regression of pseudoaneurysms has been reported [23]. In this respect, pseudoaneurysms resemble RPOC with vascularization. Although differentiation is expected to be possible by careful observation of blood flow with ultrasound, it may sometimes be difficult to differentiate these two conditions when they disappear naturally. There is a possibility that these two conditions coexist, and there may be conditions overlapping between them [14]. A pseudoaneurysm may also develop after miscarriage or delivery from the same source as RPOC, with the formation of arteriovenous fistulas and continuing development of arteriovenous communication [10]. In this process, the blood vessel may form a pool, developing into a form with characteristics different from those of RPOC. Since spontaneous regression occurs in the majority of cases, it is regarded as a condition having the same cause as RPOC and in some cases may coexist with RPOC, but since most patients are treated conservatively, there is no pathological evidence for this.

Disorders involving postpartum or post-miscarriage intrauterine vascularization also include villous-derived malignancies (e.g., invasive moles and placental site trophoblastic tumors (PSTTs)). Invasive moles and PSTTs may be differentially diagnosed by abnormally high levels of human chorionic gonadotropin (hCG) and of human placental lactogen (hPL) [26], respectively.

## 8. Standardization of Diagnosis

Generally, in practice, the treatment plan would be decided according to the degree of bleeding. Although hysterectomy is inevitable when life-threatening bleeding occurs, mild bleeding allows wait-and-see therapy. For women in the reproductive age group or women who want to preserve fertility, a variety of clinical management approaches to preserve an intact uterus have been attempted. This requires that vascular lesions be correctly evaluated. There is a need to establish uniform diagnostic definitions of the neo-vascular lesions seen after delivery or miscarriage.

New diagnostic criteria for assessing changes after the onset of RPOC are desirable. Establishing specific diagnostic definitions for each of the stages of secondary changes according to the time course of their evolution will allow assessment of the degree of risk at each stage, making it possible to respond accordingly.

AVMs are likely to have already been established when the placenta was formed. Routine screening of the adhesion of the placenta to the myometrium should be performed at some point in pregnancy. Although placenta accreta is difficult to evaluate, new diagnostic criteria for evaluating whether abnormal vascularization has developed at this site should be established.

## 9. New Technology and Future Evolution

Classically, definitive diagnosis of each of the mentioned disorders is made pathologically by the examination of resected specimens; in other words, correct diagnoses are available only retrospectively. However, clinically, uterine preservation and fertility maintenance are often the top priority. Notwithstanding pathological diagnosis, prospective diagnosis and risk assessment should be performed to facilitate the development of specific treatments. We believe that blood-flow diagnosis is an effective tool in the management of RPOC and RPOC-related diseases.

Recently, the precision of ultrasonography for identifying blood flow has improved dramatically. There has been a report of visualization of low-velocity blood flow in small vessels that was achieved using proprietary algorithms to minimize motion artifacts and high frame rates to provide high-sensitivity Doppler imaging [27]. There has also been a report of the visualization of intramyometrial vascularization and arteriovenous malformation by applying the technology of constructing three-dimensional (3D) images from blood flow data to show a 3D representation of the vasculature [28]. Detecting the vasculature at the placental–myometrial interface and classifying vascular diseases according to hemodynamics in the remnants would be very useful clinically, and such diagnoses are becoming possible using bedside ultrasonography.

## Figures and Tables

**Figure 1 jcm-10-01084-f001:**
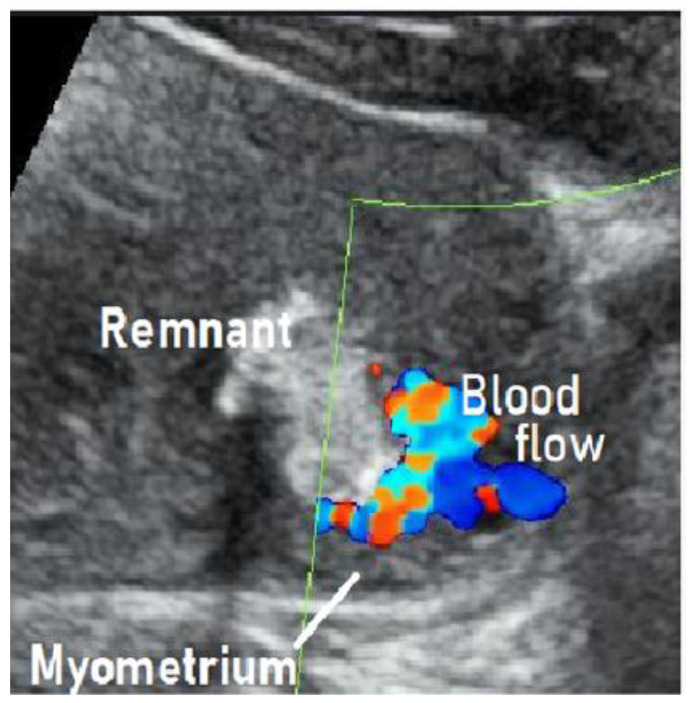
Findings of color Doppler ultrasound: Hypervascular appearance with turbulent flow. The hypervascularity was limited to the point of contact between the remnant and the myometrium, and there was no pulsatile perfusion in the remnant itself.

**Figure 2 jcm-10-01084-f002:**
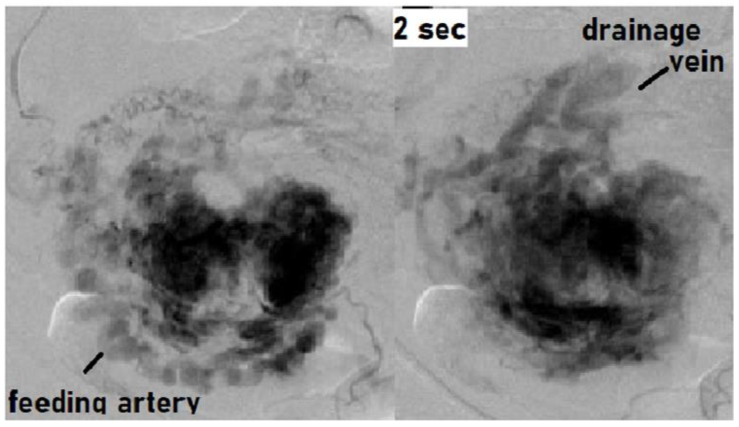
Findings of angiography: Abnormal engorgement in the arterial phase, with a thick venous phase visualized only 2 s later. This indicated the presence of an arteriovenous shunt. There was no pulsatile arterial perfusion in the remnant itself, where the contrast agent simply appeared to be leaking.

## Data Availability

This study does not report any data.

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
