# Peer review of "Overview of Neo-Vascular Lesions after Delivery or Miscarriage"

_jcm, 2021, doi:10.3390/jcm10051084_

Round 1
Reviewer 1 Report
The authors have thoroughly revised the paper and addressed its complicated structure. The current version of the paper is much more clear and readible.
I have no further comments or corrections.
Author Response
Dear Sir
Thank you very much for reviewing my manuscript.
I humbly send the certificate of English correction by native speaker.
Sincerely Yours

Reviewer 2 Report
Having reviewed this version I am pleased to see the comments taken into account. I find the structure is much improved and more readily digestible to a broader audience.
Author Response

(The authors gave the same response as above.)

Reviewer 3 Report
Thank you for the opportunity to review this review study of placental polyps. Overall this is a well written manuscript with some English language issues. I believe it is acceptable, once edited by someone for English language.
Author Response
Thank you very much for reviewing my manuscript.
I humbly send the certificate of English correction by native speaker.
Sincerely Yours

Reviewer 4 Report
The author presents one single case report of a placental remnant with underlying enhanced myometrial vascularization (EMV). They treated the case by selective uterine artery embolisation.
For most of the manuscript the author suggests a classification of highly vascularized lesions after pregnancy. The proposed classification and the reported etiology/pathogenesis are based on the authors opinion/hypotheses. Landmark papers on the subject are not cited. The theory of preexisting anteriovenous malformations (AVM) before placentation is not substantiated by facts.
I strongly disagree with the author’s point of view and classification as with the proposed management. I’d fear this paper would only add to the confusion about EMV/AVM instead of clarifying it.
References
Van den Bosch T, Van Schoubroeck D, Timmerman D. Maximum Peak Systolic Velocity and Management of Highly Vascularized Retained Products of Conception. J Ultrasound Med. 2015 Sep;34(9):1577-82. doi: 10.7863/ultra.15.14.10050. Epub 2015 Aug 7. PMID: 26254150.
Timor-Tritsch IE, Haynes MC, Monteagudo A, Khatib N, Kovács S. Ultrasound diagnosis and management of acquired uterine enhanced myometrial vascularity/arteriovenous malformations. Am J Obstet Gynecol. 2016 Jun;214(6):731.e1-731.e10. doi: 10.1016/j.ajog.2015.12.024. Epub 2016 Feb 9. PMID: 26873276.
Grewal K, Al-Memar M, Fourie H, Stalder C, Timmerman D, Bourne T. Natural history of pregnancy-related enhanced myometrial vascularity following miscarriage. Ultrasound Obstet Gynecol. 2020 May;55(5):676-682. doi: 10.1002/uog.21872. PMID: 31503383.
Van Schoubroeck D, Van den Bosch T, Scharpe K, Lu C, Van Huffel S, Timmerman D. Prospective evaluation of blood flow in the myometrium and uterine arteries in the puerperium. Ultrasound Obstet Gynecol. 2004 Apr;23(4):378-81. doi: 10.1002/uog.963. PMID: 15065189.
Author Response
Dear Sir
Thank you very much for reviewing my manuscript.
I humbly send a letter to you.
Sincerely Yours

This manuscript is a resubmission of an earlier submission. The following is a list of the peer review reports and author responses from that submission.
Round 1
Reviewer 1 Report
The paper provides an overview of neo-vascular lesions after delivery or miscarriage.
Although the paper sets out to provide an overview, the structure of the paper is complicated and the information could probably be represented in more clearly. Reading through the paper, the different subheadings seem not to combine into a nicely flowing overview. For instance section 6 and 7 describe diagnostic definitions and differential diagnosis, which would be more relevant at the start of the paper.
The following corrections could be considered:
- Line 67, mentions "first", but the "second" is missing. It is unclear what the second issue is
- Line 83, the heading mentions "treatment", while the text focusses on imaging
- Line 91, this sentence is unclear, consider rephrasing
- Line 94, consider clarifying "oral" MTX
- Line 97, consider putting 'when the level of hcg is high" at the end of the sentence
- Line 111, this sentence is unclear, consider rephrasing
- Line 122, this sentence is unclear, consider rephrasing
- Line 127, the sentence starting with "at defect" is difficult to read, consider rephrasing
- Line 135, consider "was" instead of "increased"
- Line 154, use "low" numbers instead
- Line 164, "Japanese paper"?? Pleas insert reference. Possible also add a comment on the limited number of patients. It is a very strong conclusion based on only 20 patients.
- Line 158, consider rephrasing, the numbers (1574, 71) are mentioned twice, ART-related pregnancies is a non-conventional term.
- Line 177, this sentence is unclear, consider rephrasing
Reviewer 2 Report
This is a review of a rare area of practise. Whilst the content is sound its order is somewhat confusing and for the reader it is difficult sometimes to understand how paragraphs are linked. It is also not clear about the relative timing of lesions- it is not clear if these lesions relate to development during pregnancy or are a consequence.
I think the piece would benefit significantly from being reordered and themed in a different way to help the reader understand aetiology, differential, treatment and complications. For example it is not immediately clear the link to accreta on first read.
More explanation about the paragraph introducing the difference between RPOC and placental polyps is required as this is an interesting clinical difference with different treatment options are available.
Overall I do not think this provide new insights or a particularly nuanced appraisal of the condition that would be of significant benefit to the reader. I think it would be reasonable to consider a rewrite and review.
Review order and content to make the piece flow more naturally and split the clinical conditions so the reader gets a sense of comparing and contrasting differentials with these lesions.
Reviewer 3 Report
Thank you for the opportunity to review this study examining placental polyps and RPOCs. In general, this paper is very poorly cited. There is also no flow to the paper. There are many statements made by the authors and it is unclear if these are the author's opinion or if there are studies that corroborate these statements. There are also English language issues throughout that further make the manuscript difficult to follow.
Abstract: “the villi must be completely left without damage” completely left does not make sense; please re-word for clarity
Line 106-107: please provide a reference “combination of UAE and hysteroscopy has gradually become a standard treatment.” I have never heard of this being "standard" treatment
Line 100: Please state when hysteroscopy can be safely done to remove RPOCs
Line 125-126: please provide a reference
Lines 133-135: very confusing statement; Authors state that success rates increased to 83% within 3 weeeks, but 66% within 4 (from line prior) please clarify this statement
Lines 139-141: please provide a reference for this statement
Line 143: “repeated procedures would expel the residual villi and repair the decidual defect”: couldn’t repeat procedures damage the endometrium and lead to scarring?
Lines 143-144: Kaufmann therapy is recommended for atypical bleeding that continues after a full-term delivery or miscarriage.: what is Kaufmann therapy?
Lines 156-163: please provide a reference
Lines 164-166: what Japanese paper? Please provide a reference. Also, why was UAE performed in these cases? Was it always done the same way? The reason why and how it was done is important when trying to determine its effect on fertility